# Are Lung Ultrasound Features More Severe in Children Diagnosed with Bronchiolitis after the COVID-19 Lockdown Period?

**DOI:** 10.3390/jcm11185294

**Published:** 2022-09-08

**Authors:** Danilo Buonsenso, Rosa Morello, Valentina Ferro, Anna Maria Musolino, Cristina De Rose, Riccardo Inchingolo, Piero Valentini

**Affiliations:** 1Department of Woman and Child Health and Public Health, Fondazione Policlinico Universitario Agostino Gemelli IRCCS, Università Cattolica del Sacro Cuore, 00168 Rome, Italy; 2Centro di Salute Globale, Università Cattolica del Sacro Cuore, 00168 Rome, Italy; 3Dipartimento di Emergenza e Accettazione, Ospedale Pediatrico Bambino Gesù, 00165 Rome, Italy; 4Dipartimento Scienze Mediche e Chirurgiche, UOC Pneumologia, Fondazione Policlinico Universitario Agostino Gemelli IRCCS, 00168 Rome, Italy

**Keywords:** bronchiolitis, children, COVID-19, point-of-care ultrasound, respiratory viruses, lung ultrasound

## Abstract

The non-pharmacological measures implemented during the SARS-CoV-2 pandemic disrupted the usual bronchiolitis seasonality. Some authors have speculated that, after the lock down period, there would be an increase in the number and severity of respiratory infections due to the re-introduction of respiratory viruses. We collected clinical, microbiological and lung ultrasound data using the classification of the Italian Society of Thoracic Ultrasound (ADET) in children with bronchiolitis during the pandemic compared to the pre-pandemic period, with the aim of assessing whether the epidemic of bronchiolitis during the pandemic was characterized by a more severe lung involvement documented by lung ultrasound. We enrolled 108 children with bronchiolitis (52 pre-pandemic and 56 COVID-19 period), with a median age of 1.74 months (interquartile range, IQR 1–3.68) and 39.8% were females. Rhinovirus detection and high-flow nasal cannula usage were both increased during the COVID-19 period, although overall need of hospitalization and pediatric intensive care unit admissions did not change during the two periods. Lung ultrasound scores were similar in the two cohorts evaluated. Conclusions: our study suggests that, despite changes in microbiology and treatments performed, lung ultrasound severity scores were similar, suggesting that that bronchiolitis during the pandemic period was no more severe than pre-pandemic period, despite children diagnosed during the pandemic had a higher, but it was not statistically significant, probably, due to small sample size, probability of being admitted.

## 1. Introduction

Acute bronchiolitis is one of the most substantial health problems for infants and young children worldwide [1]. It occurs in babies and children under 2 years of age and most commonly in the first year of life, peaking between 3 and 6 months [2]. Bronchiolitis is characterized by extensive inflammation and edema of the airways, increased mucus production and necrosis of airway epithelial cells [3]. Respiratory syncytial virus (RSV) is the most commonly identified virus, other associated pathogens include rhinovirus, metapneumovirus, coronavirus, human bocavirus, influenza virus, adenovirus and parainfluenza virus [4].

Bronchiolitis is a seasonal infection, with the season typically beginning in late October in the temperate northern hemisphere, peaking in January or February, and ending in April. During the COVID-19 pandemic, stay-at-home orders, physical distancing, face masks and other non-pharmaceutical interventions (NPIs) do not only impact COVID-19, but also the dynamics of various other infectious diseases. In fact, the autumn and winter bronchiolitis epidemics have virtually disappeared worldwide in the first year of the COVID-19 pandemic [5,6,7,8,9,10,11,12,13,14].

As result of the reduction of social distancing measures and the reopening of schools, viruses have started to circulate again. Indeed, all countries have experienced the resurgence of RSV following the reduction of public health measures related to the COVID-19 disease [14]. Some authors have speculated that the lack of exposure to common respiratory viruses in children born during COVID-19 lockdowns may expose them to a higher risk of severe disease if exposed later in life to traditional viruses (this theory has been called “immunity debt”) [15]. Moreover, given the lower exposure of the general (also adult) population to common viruses, major and out-of-seasons rebounds of several co-circulating respiratory viruses have been reported in several countries [16], it has also been suggested that children could have suffered more severe forms of bronchiolitis due to viral co-infections. So far, a few available studies on the topic have not documented a higher need of invasive ventilation (IV) and pediatric intensive care unit (PICU) admissions in children with bronchiolitis during the post-COVID-19 periods [17,18]. However, IV and PICU admission may only represent the very extreme of disease severity, which is relatively rare in children. Therefore, other parameters that could objectivate the degree of lung disease in children diagnosed with bronchiolitis before and after the pandemic may better show us if, overall, children have suffered a different disease severity. Clinical parameters and admissions, in fact, can be very subjective and dependent on the attending physician; therefore, they can miss the real burden of bronchiolitis before and after the pandemic. In this regard, lung ultrasound (LUS) can be a more objective parameter to define lung disease in children with bronchiolitis.

Lung ultrasound (LUS) is a feasible, portable, easy to learn and non-ionizing radiation technique. In the last decades, it has become an emerging diagnostic tool for diagnosing pneumonia and bronchiolitis in children, with remarkable sensitivity and specificity [19]. Moreover, a consensus in 2020 established the role of lung ultrasound in the diagnosis and management of pneumonia and bronchiolitis in children as an evidence-based method of imaging [20].

In our third level center, LUS was routinely performed in cases of respiratory disease, including bronchiolitis. Specifically, we are prospectively collecting LUS findings in children with bronchiolitis since 2018. In view of the enormous impact of SARS-CoV-2 infection on the bronchiolitis epidemiology and speculation about the possibility of an increase in the numbers and severity of lower respiratory tract infections after the easing of restrictions, we collected clinical and ultrasound data of patients with bronchiolitis also during the first bronchiolitis season after lockdown. Importantly, our team of clinicians performing LUS has not changed during the two periods, allowing us to compare LUS findings in two cohorts of pre- and post-pandemic children. As the ultrasound evaluation is a more objective tool to quantify the involvement of lung parenchyma during respiratory diseases [21,22], the purpose of this study was to compare the ultrasound score in bronchiolitis during COVID-19 pandemic with bronchiolitis in the pre-pandemic period to determine whether the lung ultrasound scores differ in children that had bronchiolitis in the pre- or post-pandemic period.

## 2. Materials and Methods

The study setting was a third level center in Italy (Rome). The authors prospectively collected demographic, clinical, microbiological and ultrasound data of hospitalized patient with diagnosis of bronchiolitis from 1 July 2021 to 1 January 2022. Later, these records have been compared with those prospectively collected from children with bronchiolitis evaluated in the pre-pandemic period (October 2018–March 2020). Parents/legal guardians of all included patients provided consent for study participation. The local Ethics Committee approved the study (Prot. 51720/19, ID 2921).

### 2.1. Inclusion and Exclusion Criteria

We included children younger than 12 months of age hospitalized with clinical diagnosis of bronchiolitis who underwent to thoracic ultrasound within 24 h of admission. Diagnosis of bronchiolitis was based on available guidelines and only children with a first episode were included. We excluded patients with other respiratory diseases, over 2 years of age, who have not undergone thoracic ultrasound, outside of the study period. Patients without parents/legal guardians’ consent were excluded.

### 2.2. Data Collection

Anonymous data have been collected during the study period. We obtained demographic (age, sex and gestational age), clinical parameters according to the Baraldi et al. [23] score (respiratory rate, respiratory work, temperature, oxygen saturation and feeding apnea), auscultation based by ear through stethoscope, microbiological (nasopharyngeal swabs results) and ultrasound records. In addition, we acquired information about need and degree of oxygen support (low-flow oxygen, high flow oxygen, continuous positive airway pressure (CPAP) and invasive ventilation), and other administered therapies (antibiotic therapy, bronchodilator and corticosteroid treatment). As laboratory investigations are not routinely performed in clinical practice in children with bronchiolitis, these data were not collected. Data on SARS-CoV-2 positivity have not been included since, in our Institution, we only reported SARS-CoV-2 positive bronchiolitis since January 2022 according to our prospective observations previously described [17].

### 2.3. Chest Ultrasound Examination

Esaote My Lab 40 ultrasound machine (Esaote, Genoa, Italy) with a linear (7.5 to 12 MHz) probe was used to scan the chest. The patients were scanned in supine position. The chest was divided into 14 regions according to recent method proposed by Soldati et al. [24]. Longitudinal and transversal images have been acquired. Lung ultrasound examinations were performed and classified by two pediatric residents and later reviewed by a senior pediatric doctor, with many years of experience in the field of chest ultrasound [19].

About the lung ultrasound findings, we collected the following features (as shown in Figure 1), developed according to decades of LUS studies of the Italian Academy of Thoracic Ultrasound (ADET) which described in detail the interstitial and consolidative lung syndromes [24,25,26]:-***Horizontal artifacts*** (the summation of the reverberation effects, due to the pleural-line and myofascial acoustic interfaces of the chest wall, and the mirror effects variable in its expression in relation to the thickness of the chest wall-reproducing beyond the pleural line, in a specular way, the myofascial planes of the chest wall) [25].-***Short vertical artifact*** (artifact that do not reach the bottom of the screen).-***B-line*** (hyperechogenic ultrasonographic artefacts, perpendicular to the pleural line, also known as comet-tail artefacts) that can be isolated (not more than 2 B-lines per intercostal space) or multiple (B-lines with a distance between them of less than half a cm to the confluence, remaining identifiable from each other).-***White lung*** (characterized by a granular and mostly white texture, which starts at the pleura line and ends at the bottom of the screen as reported in previously mentioned [25]).-***Sub-pleural consolidation*** (Subpleural echo-poor region interrupting the pleural line).

We classified LUS findings as follows, according to the scores agreed by the Italiana Academy of Thoracic Ultrasound, previously presented in our report standards (RED) and based on physical studies from the Academy [26]:-A-lines, normal ultrasound with score 0.-Short vertical artifact and Isolated-B lines with score 1 (counted together according to available literature See: https://doi.org/10.3390/app10051570, accessed on 24 July 2022).-Multiple B-lines with score 2.-White lung and subpleural consolidation less than 1 cm in size with score 3.-Sub-pleural consolidation greater than 1 cm in size, score 4.

### 2.4. Statistical Analysis

A statistical analysis was performed using the software STATA/IC 14.2 version 2017 (StataCorp LLC, College Station, TX, USA). We tested the normality by Skewness/Kurtosis test. Data were reported as median values with an interquartile range (IQR), and direct comparisons were made with Mann–Whitney rank-sum tests. Percentages were used to describe categorical outcomes, and distributions of categorical data were compared with either a Pearson’s χ^2^ test or a Fisher’s exact test, as appropriate. A logistic regression analysis was applied to detect predictive characteristics of the bronchiolitis during COVID-19 period was compared with pre-COVID-19 period. The inclusion of variables in the model was based on clinical plausibility and significant difference on the bivariate analysis. Considering the rarity of events or the unbalanced distribution of the cases for some variables and the small-sample bias, we performed the Firth method to reduce the small sample bias of maximum likelihood coefficients. Adjusted odds ratio (OR) and 95% confidence intervals (95% CI) were used as measures of effect. The statistical significance was set at *p* < 0.05 for all tests.

## 3. Results

### 3.1. Study Population

During the study period, 108 children with a clinical diagnosis of bronchiolitis were enrolled, 56 infants during the COVID-19 period, and 52 during the pre-COVID-19 period. Clinical and demographic characteristics of the study population, during the COVID-19 period compared to the pre-pandemic period, are described in Table 1. Overall, 60.2% of the total population were male and 39.8% female. The average age 1.7 months, 12.9% were premature. In both groups it was the first episode, rhinitis was always present, feeding difficulty was found in 69.4% of infants without significant differences between two groups. About microbiological results, a virus was identified in 107 out of 108 infants. RSV was the commonest isolate followed by rhinovirus, metapneumovirus, parainfluenza and adenovirus/bocavirus (as shown in Table 1). RSV and rhinovirus were found, respectively, to be 80.3% and 39.2% during the pandemic period and 59.6% and 1.9% during the pre-pandemic period. Management and therapies were showed in Table 1. Overall, 83.3% of infants needed oxygen support, 55.3% needed high-flow nasal cannula (HFNC) and 10.7% needed CPAP during the pandemic period, compared to 32.6% who needed HFNC and no one needing CPAP during the pre-pandemic period. No one needed to mechanical ventilation in both groups. The most common therapies administered were antibiotic, bronchodilator, corticosteroid (36.1%, 23.15% and 32.4%, respectively). They were more used in the pre-pandemic compared to the pandemic period.

### 3.2. LUS Features

Abnormal ultrasound findings were detected in all patients. B-lines were the sonographic artifacts mainly observed in most pulmonary fields explored. Instead, white lung area and subpleural consolidations (especially less than 1 cm) were found in posterior and paravertebral fields. The average score for the anterior and lateral lung fields was 1, the mean score of posterior field was 3, with similar rates in the right and left lung. No pleural effusion was detected. Interstitial and alveolar-interstitial syndromes were most frequently observed in inferior and posterior, as well as paraspinal portions of the lung. The average score in total population was 12; the mean value in the COVID-19 period group was 13 compared to 11 in the pre-pandemic group. Overall, there were not statistically significant differences in lung ultrasound findings between the two groups as shown in Table 1. Logistic regression models found that the only significant changes during the study periods were a younger age in the pre-pandemic cohort, higher rates of rhinovirus and lower detection of wheezing in the post pandemic cohort (Table 2).

## 4. Discussion

In this study, we addressed LUS severity in children diagnosed with bronchiolitis before and after the COVID-19 pandemic. Such a research was possible because in our center there is an ongoing study addressing LUS in all children evaluated in our Institution and, since it started in 2018, the same three operators were still working in the same Institution. The rational of this study was related to the indirect impact that the SARS-CoV-2 pandemic had on seasonality of bronchiolitis, with the first winter seasons after COVID-19 seeing almost no cases [5,6,7,8,9,10,11,12,13] while, during the second pandemic year, seasonality was disrupted [17,18]. These changes led some authors to speculate that the “immune debt” of children missing bronchiolitis during the first two years of life might have predisposed to more severe disease [15]. Moreover, as the general population has been less exposed to viruses for two years, this may have led to a major out-of-season re-circulation of viral infections and, consequently, a different burden of diseases [16]. A standardized LUS approach performed according to the pediatric Italian Thoracic Ultrasound Society (ADET) allowed us to compare the two cohorts. Overall, we found that the severity of lung involvement investigated through LUS was similar in children diagnosed with bronchiolitis before and after the pandemic. This is an important finding in our opinion, considering that rate of admissions or use of non-invasive ventilation can be not objective data of disease severity, as they can change according to the ability of clinicians or non-PICU settings to use these methodologies, or even to the ease of access to the HFNC, whose use is increasing worldwide despite there is no clear evidence of its efficacy in slowing progression to severe disease. In fact, in our study, were found a greater need for respiratory assistance (HFNC and CPAP), an increase in respiratory viruses and a reduced use of therapies. However, an explanation for these findings could be a different clinical management between two periods. For example, in recent years, for respiratory assistance, there has been a greater spread in the use of high flow in respiratory diseases and therefore, also, in bronchiolitis. Moreover, during the pandemic period increased attention was paid to nasopharyngeal swabs and etiological identification of respiratory infections. Finally, in the last years there has been a greater adherence to the guidelines on the management of bronchiolitis and a consequent lesser use of not recommended therapies. Therefore, in our study the parameter “need for mechanical ventilation” could be used as a more objective indicator of the more severe spectrum of bronchiolitis, however in both study periods no one in our settings required it. Therefore, LUS can be a more reproducible tool to quantify lung involvement of children with less severe bronchiolitis [19,22,23,26].

More in detail, the most common ultrasound abnormalities found in our patients with bronchiolitis were B-lines, small subpleural consolidations and atelectasis. These findings are the result of the pathogenetic process in the respiratory tract, similarly to what was reported from pre-pandemic literature [25]. Vertical artifacts are found in various pathophysiological conditions that alter the peripheral lung layers. To date, their genesis is quite complex and not completely understood. In practical terms, the sequence of events producing vertical artifacts arise from a diseased pleura covering a hyperdense non-consolidated lung.

Other common finding is the presence of multiple small subpleural consolidations with sizes of less than 1 cm with or without adjacent single or confluent B-lines. Indeed, in patients with more severe clinical picture were detected consolidations of larger dimensions (usually more than 1 cm) with static aerial bronchograms. Their presence is usually associated with the identification of atelectasis. In bronchiolitis posterior and paravertebral zones are more frequently affected. An explanation to this finding could be that children are in preferential position supine for most of the time, and therefore, the secretions follow the force of gravity. However, analysis of the results showed that ultrasound score during the COVID-19 period was overlapping with ultrasound score in bronchiolitis during the pre-pandemic period; there would not be differences as confirmed also by the logistic regression (Table 2). Therefore, ultrasound images in bronchiolitis are not more serious after the SARS-CoV-2 pandemic. These results confirm the clinical findings that would disprove a greater severity of bronchiolitis with the recirculation of respiratory viruses. In addition, this paper adds to the vast amount of previous studies [17,27,28,29,30,31,32,33,34,35,36,37,38,39] that LUS is a useful tool to support in the assessment of functional status and severity of the disease, along with routine, clinical data, of children with bronchiolitis. Specifically, can help quantify aeration of the peripheral lung, monitor consolidation up to document possible bacterial superinfection [40,41,42,43], detect pneumothorax [44] or effusions that may guide different management [19].

With regard to strengths and limitations, the ultrasound examination is an operator-dependent technique; however, in this study LUS was conducted by the same three operators working together for about 10 years, having more over a standardized approach and LUS features classification. Moreover, a limit of our study was a relatively small sample, which missed the most severe spectrum of bronchiolitis requiring invasive ventilation. Importantly, we have to highlight that our retrospective study may lack of power as a potential explanation for our findings, as according to the estimates from the logistic regression analysis reported in Table 2 children during the post COVID-19 period had a higher frequency of hospitalization and ICU admission, although statistically non-significant. Therefore, our study needs to be interpreted cautiously and needs confirmation from larger numbers. Moreover, due to the initial study design, which was aimed at detecting clinical and ultrasound data, nature of this study, we did not collect vaccination history in all children although mandatory vaccinations are performed in more than 95% of infants in children since age of 60 days (https://www.epicentro.iss.it/vaccini/coperture-infanzia-adolescenza-2019, accessed on 24 July 2022), nor breastfeeding/&formula feeding.

In conclusion, our LUS study support preliminary evidence of an overall similar LUS scores in children diagnosed with bronchiolitis during the COVID-19 pandemic, compared with a pre-pandemic cohort, but children during the pandemic period had a trend toward higher frequency of hospitalization. Importantly, similar studies with larger sample sizes, including LUS features, should be continued as epidemiologic scenarios are evolving and it will remain important to monitor the evolution of bronchiolitis during the post-pandemic era.

## Figures and Tables

**Figure 1 jcm-11-05294-f001:**
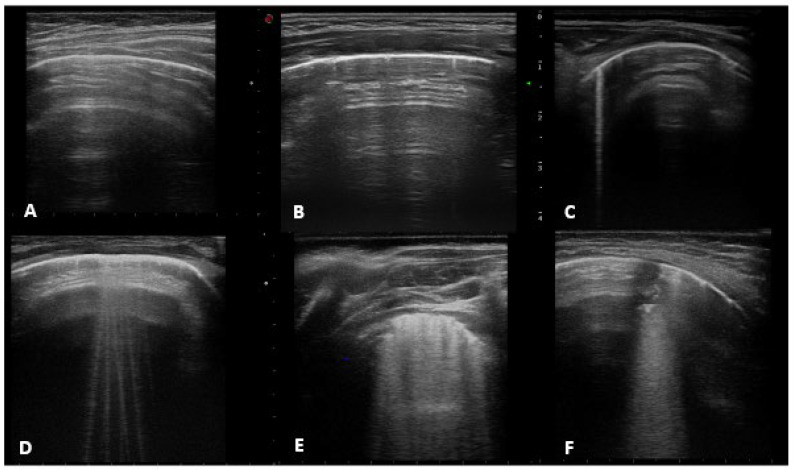
LUS findings. (**A**) A-lines, normal ultrasound; (**B**) short vertical artifact; (**C**) isolated-B line; (**D**) multiple B lines; (**E**) white lung; (**F**) subpleural consolidation.

**Table 1 jcm-11-05294-t001:** Basic characteristics of the study population.

Characteristics	Total Population *n* = 108	Pre-COVID-19 Period *n* = 52	COVID-19 Period *n* = 56	*p* Value
**Sex, *n*(%)**				
**Male**	65 (60.19)	36 (69.23)	29 (51.79)	0.06
**Female**	43 (39.81)	16 (30.77)	27 (48.21)	
**Age (months), median (IQR)**	1.74 (1–3.68)	1.6 (0.97–3.45)	2.1 (1.02–4.86)	0.22
**Prematurity, *n* (%)**	14 (12.96)	5 (9.62)	9 (16.07)	0.32
**Onset of symptoms (hours), median (IQR)**	72 (48–96)	48.5 (48–96)	72 (36–96)	0.75
**First episode, *n* (%)**	104 (96.30)	50 (96.15)	54 (96.43)	0.94
**Coryza, *n* (%)**	108 (100%)	56 (100%)	52 (100%)	-
**Difficulty of feeding, *n* (%)**	75 (69.44)	39 (75.00)	36 (64.29)	0.23
**On ongoing therapy, *n* (%)**	36 (33.33)	17 (32.69)	19 (33.93)	0.89
**Crackles on thorax auscultation, *n* (%)**	104 (96.30)	48 (92.31)	56 (100.00)	0.034
**Wheeze on thorax auscultation, *n* (%)**	32 (29.63)	28 (53.85)	4 (7.14)	<0.001
**Chest retractions, *n* (%)**	103 (95.37)	48 (92.31)	55 (98.21)	0.144
**Fever, *n* (%)**	41 (37.96)	17 (32.69)	24 (42.86)	0.277
**SaO2 < 91%, *n* (%)**	39 (36.11)	21 (40.38)	18 (32.14)	0.37
**RSV detection, *n* (%)**	76 (70.37)	31 (59.62)	45 (80.36)	0.018
**Rhinovirus detection, *n* (%)**	23 (21.30)	1 (1.92)	22 (39.29)	<0.001
**Metapneumovirus, *n* (%)**	4 (3.70)	1 (1.92)	3 (5.36)	0.34
**Parainfluenza, *n* (%)**	2 (1.85)	0	2 (3.57)	0.27
**Adenovirus/Bocavirus, *n* (%)**	2 (1.85)	0	2 (3.57)	0.27
**LUS*-right anterior side, median, (IQR)**	1 (1–3)	1 (1–3)	1.5 (1–3)	0.94
**LUS*-left anterior side, median, (IQR)**	1 (1–2.5)	1 (1–2)	1 (1–3)	0.95
**LUS*-right lateral side, median, (IQR)**	1 (1–3)	1 (1–2.5)	1 (1–3)	0.37
**LUS*-left lateral side, median, (IQR)**	1 (1–3)	1 (1–3)	1 (1–3)	0.11
**LUS*-right posterior side, median, (IQR)**	3 (2–3)	3 (1–3)	3 (2–3)	0.37
**LUS*-left posterior side, median, (IQR)**	3 (1–3)	2 (1–3)	3 (1–3)	0.34
**LUS*-paravertebral side, median, (IQR)**	1 (1–1)	1 (1–1)	1 (1–1)	0.48
**Total LUS*, median, (IQR)**	12 (9.16)	11 (9–15)	13 (8.5–17)	0.1
**Need for oxygen support, *n* (%)**	90 (83.33)	39 (75.00)	51 (91.07)	0.025
**Need for HFNC, *n* (%)**	48 (44.44)	17 (32.69)	31 (55.36)	0.018
**Need d for N-CPAP, *n* (%)**	6 (5.56)	0	6 (10.71)	0.015
**Need for mechanical ventilation, *n* (%)**	0	0	0	0
**Need for antibiotic therapy, *n* (%)**	39 (36.11)	24 (46.15)	15 (26.79)	0.036
**Trial of bronchodilator, *n* (%)**	25 (23.15)	16 (30.77)	9 (16.7)	0.07
**Need for corticosteroid treatment, *n* (%)**	35 (32.41)	24 (46.15)	11 (19.64)	0.003

IQR: interquartile range; RSV: respiratory syncitial virus; LUS*: lung ultrasound score; HFNC: high flow nasal cannulae; N-CPAP: continuous positive airway preassure.

**Table 2 jcm-11-05294-t002:** Firth logistic regression model exploring the predictors associated with COVID-19 period.

COVID-19 Period vs. Pre COVID-19 Period	Coeff.	OR	Std. Err.	z	*p* > |z|	95% CI	
**Age (months)**	0.33	1.4	0.15	2.23	0.026	0.04	0.6
**Sex** **male vs. female**	−0.62	0.54	0.69	−0.9	0.369	−2.0	0.7
**Crackles on thorax auscultation**	2.11	8.27	2.16	0.98	0.327	−2.1	6.3
**Wheeze on thorax auscultation**	−4.26	0.01	1.48	−2.88	0.004	−7.2	−1.4
**RSV detection**	0.90	2.46	1.06	0.85	0.39	−1.2	3.0
**Rhinovirus detection**	3.16	23.5	1.62	1.95	0.05	−0.02	6.3
**Need for antibiotic therapy**	−0.82	0.44	0.83	−0.99	0.32	−2.4	0.8
**Need for corticosteroid treatment**	−1.03	0.35	0.75	−1.37	0.17	−2.5	0.4
**Need for oxygen support**	−0.75	0.47	1.33	−0.56	0.57	−3.4	1.9
**Need for HFNC**	0.59	1.8	0.74	0.80	0.42	−0.86	2.04
**Need for hospitalization**	3.41	30.4	2.41	1.42	0.16	−1.31	8.13
**Need for Intensive Care Unit**	2.66	14.3	2.28	1.17	0.24	−1.80	7.12
**Constant**	−5.19	0.005	2.80	−1.86	0.06	−10.67	0.28

Coeff. = the coefficient. These estimates tell you about the relationship between the independent variables and the dependent variable, where the dependent variable is on the logit scale. OR = Odds ratio. These values are defined as exp^(b)^ and means that we exponentiate the coefficient. Std. Err. = the standard errors associated with the coefficients. z and *p* > |z|—these columns provide the z-value and 2-tailed *p*-value used in testing the null hypothesis that the coefficient (parameter) is 0.95% CI = This shows a 95% confidence interval for the coefficient.

## Data Availability

Not applicable.

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
