# Peer review of "Are Lung Ultrasound Features More Severe in Children Diagnosed with Bronchiolitis after the COVID-19 Lockdown Period?"

_jcm, 2022, doi:10.3390/jcm11185294_

Round 1

Reviewer 1 Report

This is a very interesting article on a hot topic of changing epidemiology of bronchiolitis post-Covid pandemics as well as non-invasive methods of lung monitoring.

What is lacking is an explanation of an acronym ADET (in Discussion part). It was not mentioned in the Methods or Results sections. Is it the same as "scores agreed by the Italiana Academy of Thoracic Ultrasound"?

Author Response

Thank you very much for appreciating our work, we have clarified the ADET meaning in the discussion, which was referred to the Italian Academy of Thoracic Ultrasound as you mentioned

Reviewer 2 Report

.- Topic of great interest for potential readers of this Journal.

.- Some minor comments are made in favor of improving the current version of the manuscript:

Summary. See the sections on introduction, methodology, results, discussion and conclusions are missing. Perhaps the section on methodology and results could be expanded. Results are confused with conclusions. Better separate.

Keywords. Maybe add “respiratory viruses”. Consider changing the acronym “LUS”, which has not been previously referred to, to “lung ultrasound”.

Introduction. Maybe it will be long. It should be noted that of the 25 references listed in the manuscript, 21 are included in this section.

Methodology. Chest ultrasound examination, could summarize some of its sections that refer to basic concepts of pulmonary ultrasound.

Results. Study population. Perhaps the first paragraph could be summarized by referring to the data presented in Table 1.

Tables. Table 1. Consider changing the title to “basic characteristics of the study population”.

Discussion. Well thought out and exposed, adding a section on strengths and limitations.

Conclusions. Perhaps it could be a separate paragraph after strengths and limitations.

References. Correct and well exposed. It is very appreciated that of the 25 references, 22 are recent (equal to or less than 5 years). Congratulations!!

Author Response

Thank you very much for your support. Please find below a point-by-point response to your comments. Changes have been highlighted in tracked changes in the main document.

.- Topic of great interest for potential readers of this Journal.

Thank you very much

.- Some minor comments are made in favor of improving the current version of the manuscript:

Summary. See the sections on introduction, methodology, results, discussion and conclusions are missing. Perhaps the section on methodology and results could be expanded. Results are confused with conclusions. Better separate.

Thank you for this suggestion, we fully agree. We improved the results section and separated the conclusions. Conversely, the introduction, methodology, results, discussion because of the journal guidelines.

Keywords. Maybe add “respiratory viruses”. Consider changing the acronym “LUS”, which has not been previously referred to, to “lung ultrasound”.

Thanks again, we changed as suggested

Introduction. Maybe it will be long. It should be noted that of the 25 references listed in the manuscript, 21 are included in this section.

Thank you, we have expanded the discussion with several references

MethodologyChest ultrasound examination, could summarize some of its sections that refer to basic concepts of pulmonary ultrasound.

Thank you, we added a sentence with reference to refer to further details.

ResultsStudy population. Perhaps the first paragraph could be summarized by referring to the data presented in Table 1.

Thanks, although we agree with you, as the results section is already a bit short, we would appreciate to leave as it is now

Tables. Table 1. Consider changing the title to “basic characteristics of the study population”.

Thanks, changed as suggested

Discussion. Well thought out and exposed, adding a section on strengths and limitations.

Thanks for appreciating

Conclusions. Perhaps it could be a separate paragraph after strengths and limitations.

Thanks for spotting this, we separated it now from the previous paragraph

References. Correct and well exposed. It is very appreciated that of the 25 references, 22 are recent (equal to or less than 5 years). Congratulations!!

Thanks a lot for your support

Reviewer 3 Report

Respiratory viral infections represent a significant clinical problem, especially in pediatric practice. In this regard, the use of new methods of diagnosis and severity assessment is of practical interest. The analysis of lung ultrasound data in children with bronchiolitis in this article is interesting.

Comments:

1.      The rationale for the purpose of the study and the supposed relationship to "immunity debt" is not clear, since the children in the study were an average age of 1.7 months, and they have no history of previous disease.

2.      There is very little information in the materials and methods about the instruments that were used to obtain clinical and laboratory data. What clinical criteria were used to assess functional status? Was auscultation analysis by ear or was there digital analysis (these data have statistical differences, so it is important that they were obtained in the same way)? Was Sars-Cov2 detected in patients? Were antibodies determined? Were the children breastfed or formula-fed? Have the children been vaccinated against any infections? It is recommended to add more clinical information, lab results.

3.      Table 1 has some typos, it is recommended to check the data, especially the sex of the children, there are mistakes there.

4.      It is recommended to add research limitations.

5.      It is recommended to add clinical significance have the results of the study, as well as to add future research perspectives. The diagnostic capabilities of the ultrasound study are of interest, including its potential in the choice of management tactics, assessment of functional status and severity of the disease.

Author Response

Thank you very much for your support. Please find below a point-by-point response to your comments. Changes have been highlighted in tracked changes in the main document.

Respiratory viral infections represent a significant clinical problem, especially in pediatric practice. In this regard, the use of new methods of diagnosis and severity assessment is of practical interest. The analysis of lung ultrasound data in children with bronchiolitis in this article is interesting.

Thanks

Comments:

  1. The rationale for the purpose of the study and the supposed relationship to "immunity debt" is not clear, since the children in the study were an average age of 1.7 months, and they have no history of previous disease.

Thank you, we have clarified that both the immunity debt can have directly impacted on children according to literature (although as you noted this cohort is young) but also on the general circulation of viruses as also adutls have been less exposed, potentially contributing to out of season outbreaks, and therefore this might theoretically have led to different disease course.

  1. There is very little information in the materials and methods about the instruments that were used to obtain clinical and laboratory data. What clinical criteria were used to assess functional status? Was auscultation analysis by ear or was there digital analysis (these data have statistical differences, so it is important that they were obtained in the same way)? Was Sars-Cov2 detected in patients? Were antibodies determined? Were the children breastfed or formula-fed? Have the children been vaccinated against any infections? It is recommended to add more clinical information, lab results.

Thank you very much for your comments. WE have clarified in the methods the clinical and auscultation information, we have clarified that we had no covid cases in these cohorts, and that since we do not routinely perform lab tests in bronchiolitis, these data were not collected. Similarly, we did not collect vaccination history in all children although mandatory vaccinations are performed in more than 95% of infants in children since age of 60 days, nor breastfeeding/&formula feeding, and we have added this in the study limitation.

  1. Table 1 has some typos, it is recommended to check the data, especially the sex of the children, there are mistakes there.

Thank you, we have corrected it

  1. It is recommended to add research limitations.

Thanks, limitaions have been further implemented based on your comments

  1. It is recommended to add clinical significance have the results of the study, as well as to add future research perspectives. The diagnostic capabilities of the ultrasound study are of interest, including its potential in the choice of management tactics, assessment of functional status and severity of the disease.

Thank you, we have implemented this section in the discussion

Round 2

Reviewer 3 Report

The authors have made changes to the manuscript that have improved its quality, but questions still remain:

1.      The abstract states that a total of 108 children with bronchiolitis were examined (56 pre-pandemic and 52 post-pandemic period), but the main body of the article states that 56 infants were examined during the Covid-19 period and 52 during the pre-Covid-19 period. The numbers are mixed up and there is a mismatch in the periods (the «post pandemic period» that is in the abstract is not in the results). It is recommended to make the terminology the same in different parts of the manuscript and correct the numbers.

2.      In Table 1, there is a typo in the line sex - female in the pre-Covid-19 period column. There is an extra digit 27.

3.      The conclusions in the abstract and the article are different:

«Conclusions: our study suggests that, despite changes in microbiology and treatments performed, lung ultrasound severity scores were similar, suggesting that that bronchiolitis during the pandemic period was no more severe than pre-pandemic period».

«In conclusion, our LUS study supports preliminary evidence of an overall more severe disease of children diagnosed with bronchiolitis after the Covid-19 pandemic, compared with a prepandemic cohort, despite the important epidemiologic changes occurred during the last two years».

It is recommended to modify the conclusion according to the results.

Author Response

Thanks a lot for your very useful comments, please find attached our replies which we have highlighted in the new veersion of the manuscript.

The authors have made changes to the manuscript that have improved its quality, but questions still remain:

  1. The abstract states that a total of 108 children with bronchiolitis were examined (56 pre-pandemic and 52 post-pandemic period), but the main body of the article states that 56 infants were examined during the Covid-19 period and 52 during the pre-Covid-19 period. The numbers are mixed up and there is a mismatch in the periods (the «post pandemic period» that is in the abstract is not in the results). It is recommended to make the terminology the same in different parts of the manuscript and correct the numbers.

During the revision we noted that in the first row of the tab we mischanged the two periods, we have corrected it in the table but I forgot to change in the abstract, sorry and thanks for noticing it. We have not correct accordingly.

  1. In Table 1, there is a typo in the line sex - female in the pre-Covid-19 period column. There is an extra digit 27

Thanks, correct now

  1. The conclusions in the abstract and the article are different:

«Conclusions: our study suggests that, despite changes in microbiology and treatments performed, lung ultrasound severity scores were similar, suggesting that that bronchiolitis during the pandemic period was no more severe than pre-pandemic period».

«In conclusion, our LUS study supports preliminary evidence of an overall more severe disease of children diagnosed with bronchiolitis after the Covid-19 pandemic, compared with a prepandemic cohort, despite the important epidemiologic changes occurred during the last two years».

It is recommended to modify the conclusion according to the results.

Thanks again, it was a typo in the conclusion, we meant NOT MORE SEVERE, sorry we have corrected it now.